# Aging-Related Changes in the Ultrastructure of Hepatocytes and Cardiomyocytes of Elderly Mice Are Enhanced in ApoE-Deficient Animals

**DOI:** 10.3390/cells10030502

**Published:** 2021-02-26

**Authors:** Małgorzata Łysek-Gładysińska, Anna Wieczorek, Artur Jóźwik, Anna Walaszczyk, Karol Jelonek, Grażyna Szczukiewicz-Markowska, Olaf K. Horbańczuk, Monika Pietrowska, Piotr Widłak, Dorota Gabryś

**Affiliations:** 1Division of Medical Biology, Institute of Biology, University of Jan Kochanowski, Uniwersytecka 7, 25-406 Kielce, Poland; anna.wieczorek@ujk.edu.pl; 2Institute of Genetics and Animal Biotechnology PAS, Jastrzębiec, Postępu 36A, 05-552 Magdalenka, Poland; 3Biosciences Institute, Newcastle University, Newcastle upon Tyne NE1 7RU, UK; walaszczyk.anna@gmail.com; 4Center for Translational Research and Molecular Biology of Cancer, Maria Sklodowska-Curie National Research Institute of Oncology Gliwice Branch, Wybrzeże Armii Krajowej 15, 44-101 Gliwice, Poland; Karol.Jelonek@io.gliwice.pl (K.J.); Monika.Pietrowska@io.gliwice.pl (M.P.); Piotr.Widlak@io.gliwice.pl (P.W.); 5Department of Surgical Medicine with the Laboratory of Medical Genetics, Collegium Medicum, University of Jan Kochanowski, al. IX Wieków Kielc 19A, 25-317 Kielce, Poland; grazyna.szczukiewicz-markowska@ujk.edu.pl; 6Faculty of Human Nutrition, Warsaw University of Life Sciences, Nowoursynowska 159 C, 02-776 Warsaw, Poland; olaf_horbanczuk@sggw.edu.pl; 7Department of Radiotherapy, Maria Sklodowska-Curie National Research Institute of Oncology, Gliwice Branch, Wybrzeże Armii Krajowej 15, 44-101 Gliwice, Poland; dorota.gabrys@io.gliwice.pl

**Keywords:** aging, hepatocytes, cardiomyocytes, lysosomes, mitochondrion, lipids

## Abstract

Biological aging is associated with various morphological and functional changes, yet the mechanisms of these phenomena remain unclear in many tissues and organs. Hyperlipidemia is among the factors putatively involved in the aging of the liver and heart. Here, we analyzed morphological, ultrastructural, and biochemical features in adult (7-month-old) and elderly (17-month-old) mice, and then compared age-related features between wild type (C57Bl/6 strain) and ApoE-deficient (transgenic ApoE^−/−^) animals. Increased numbers of damaged mitochondria, lysosomes, and lipid depositions were observed in the hepatocytes of elderly animals. Importantly, these aging-related changes were significantly stronger in hepatocytes from ApoE-deficient animals. An increased number of damaged mitochondria was observed in the cardiomyocytes of elderly animals. However, the difference between wild type and ApoE-deficient mice was expressed in the larger size of mitochondria detected in the transgenic animals. Moreover, a few aging-related differences were noted between wild type and ApoE-deficient mice at the level of plasma biochemical markers. Levels of cholesterol and HDL increased in the plasma of elderly ApoE^−/−^ mice and were markedly higher than in the plasma of elderly wild type animals. On the other hand, the activity of alanine transaminase (ALT) decreased in the plasma of elderly ApoE^−/−^ mice and was markedly lower than in the plasma of elderly wild type animals.

## 1. Introduction

Aging is characterized by the progressive loss of homeostasis at genomic, cellular, tissue, and whole-organism levels leading to a decreased ability to respond to stress, functional decline, and increased risk of morbidity and mortality [1,2]. These changes are a major risk factor for serious pathologies, including cancer, diabetes, cardiovascular disorders, and neurodegenerative diseases [2,3]. The accumulation of damaged macromolecules and organelles is one of the most persistent changes in aging cells and is associated with the decline of different catabolic pathways. At the cellular level, oxidative damage is characterized by disrupted mitochondria, lysosomes, endoplasmic reticulum [4]. Mitochondria are responsible for ATP production and regulate many age-related pathways including senescence, autophagy, and inflammation [5,6]. Mitochondria are considered the main intracellular source of superoxide anion (O_2_^−^), as well as the major target of free radical attacks. ROS produced by the mitochondrial respiratory chain damage mitochondrial constituents, including proteins, lipids, and mitochondrial DNA [7,8,9,10,11].

The signs of aging depend on the type of organ, reflecting its anatomy and functions. The liver is a complex metabolic organ with a wide range of functions, including detoxification, protein synthesis, and the production of compounds necessary for digestion [12]. Certain age-related changes in the senescent liver contribute to systemic susceptibility to age-related diseases [13,14]. This process is promoted by alterations in the genome and epigenome that contribute to the dysregulation of mitochondrial function and nutrient-sensing pathways. Cellular senescence and low-grade inflammation impact directly on multiple phenotypic changes in different types of liver cells, i.e., hepatocytes, liver sinusoidal endothelial, hepatic stellate, and Küpffer cells [12,15]. Various age-related hepatic changes have been reported, such as increased hepatocyte size, increases in the number of binucleated cells, and a reduction in mitochondrial number [16,17,18]. These changes may significantly affect liver morphology, physiology, and oxidative capacity. Similarly, biological aging promotes progressive structural alterations of the heart and vasculature. These changes include vascular stiffening, increased left ventricular wall thickness, fibrosis with aging, as well as some functional changes and compensatory responses that the aged heart undergoes which diminish its ability to respond to increased workloads and decreased reserve capacity [19]. Aging-related changes in cardiac mitochondria include changes in their content and morphology, the activity of the complexes of the electron transport chain (ETC), the opening of the mitochondrial permeability transition pore (MPTP), ROS formation, and mitochondrial dynamics. Since the incidence of cardiovascular disease increases with age, impaired mitochondrial function is associated with the increased vulnerability of individuals to developing various age-related cardiovascular diseases [20,21].

Aging processes are accelerated by many factors such as smoking, ultraviolet light, and other environmental factors [22,23,24]. Additionally, obesity, atherosclerosis, and dyslipidemia produce a variety of alterations that may cause changes in both liver and cardiac morphology which predispose to organ dysfunction [8,25]. ApoE^−/−^ mice are a practical model of hyperlipidemia and atherosclerosis. This is because these mice show delayed lipoprotein clearance and consequently develop hyper- and dyslipoproteinemia, severe hypercholesterolemia, and atherosclerotic lesions even when on a normal diet [26]. Here, we aimed to address the hypothetical role of hyperlipidemia on heart and liver aging. Morphological, ultrastructural, and biochemical features were analyzed in adult (7 months old) and elderly (17 months old) mice then age-related features were compared between wild type C57Bl/6 and transgenic ApoE^−/−^ animals.

## 2. Materials and Methods

*Animals.* Male C57Bl/6J mice, either wild type (wt) or homozygous ApoE deficient (ApoE^−/−^) were purchased from Charles River Laboratories (Research Models and Services, Sulzfeld, Germany GmbH); the different genotype groups were not littermates. In each group, we included 10 animals. The animals were kept at room temperature (21 °C) in a naturally controlled 12:12 ratio of light and dark and were given laboratory chow ad libitum. The mice were kept on a regular diet. At the age of 7 months (adult) and 17 months (elderly), the animals were sacrificed by cervical dislocation, and liver and heart tissues were immediately taken for further analyses. All the procedures were conducted in conformance with institutional guidelines and in compliance with national and international laws and policies, the study was approved by the appropriate Local Ethics Committee for Experimentation with Animals, Katowice, Poland No 72/2012.

*Analysis of ultrastructure*. Immediately after mice decapitation, resection of the distal part of the left liver lobe and a fragment of the ventricular myocardium was performed. Both tissues were cut into proper size pieces (2 mm^3^) and fixed by immersion in buffered 3% glutaraldehyde in cacodylate buffer (pH 7.2) for at least 2 h at 4 °C. The tissue specimens were then post-fixed in 2% osmium tetroxide in cacodylate buffer (pH 7.2) for 1 h at 4 °C. Dehydration of the fixed tissues was performed using an ascending series of ethanol and then transferred into epoxy resin via propylene oxide [27]. Finally, the liver and heart samples were embedded in a mixture of DDSA/NMA/Embed-812 (Agar Scientific Ltd., Stansted, UK). Ultra-thin sections (40–60 nm) were cut on a Reichert-Jung ultramicrotome and double-stained with uranyl acetate and lead citrate. Evaluation of ultrastructure was performed using a transmission electron microscope Tesla BS-500 with Frame Transfer-1K-CCD-Camera (TRS, Munich, Germany). Four mice from each test group were used to evaluate ultrastructural analysis. Further, 12 epon blocks were prepared for each group of mice. Three grids were analyzed from each resin block. Morphometric analysis of mitochondria and lysosomes were performed in 100 random electron micrographs from each animal group. The average numbers of organelles per cell were calculated for each animal group. The length (µm) of mitochondria was also measured. The results were subjected to statistical treatment with the use of the XLSTAT add-in for Microsoft Excel. The conducted multi-way (age, genetic groups) analysis of Kruskal–Wallis rank test (nonparametric alternative to the ANOVA) enabled the identification of significant effects of differences at the level of the *p*-value. In addition, multiple pairwise comparisons were performed to analyze the difference between each pair using Dunn’s test; *p* < 0.05 was considered the significance threshold.

*Analysis of histomorphology*. For analysis of histomorphology, liver and heart semi-thin sections (500 nm) were prepared using a Reichert-Jung ultramicrotome (Reichert, Vienna, Austria), stained with toluidine blue, and examined using a light microscope Zeiss Axio Scope (Oberkochen, Germany). A1 with a black and white digital camera. Areas of lipid deposits were determined with Photoshop CS6 software (Adobe), using the sample color area tool for black (R = 0; G = 0; B = 0) and then recording the number of pixels in a histogram window. Pixels were counted for 3 random pictures in each group. A total of 10 specimens of 10 semi-thin sections were prepared and 100 pictures were taken in each group of mice. The results were subjected to statistical treatment with the use of the XLSTAT add-in for Microsoft Excel. The conducted multi-way (age, genetic groups) analysis of variance ANOVA enabled the identification of significant effects of differences at the level of *p*-value. In addition, multiple pairwise comparisons were performed to analyze the difference between each pair using Dunn-Sidak’s test; *p* < 0.05 was considered a significance threshold.

*Analysis of lysosomal beta-galactosidase*. Immediately after resection, the liver tissue was homogenized in a medium consisting of 0.25 M sucrose (1 g tissue per 7 mL sucrose) in a Potter–Elvehjem glass homogenizer with a Teflon piston operated at 200 rpm according to the modified method of Marzella and Glaumann [28]. The homogenates were fractionated by differential centrifugation. The first centrifugation was performed at 1000× *g* for 10 min to remove cell debris, nuclei, and heavy mitochondria. The resulting supernatant was centrifuged at 20,000× *g* for 20 min; the resulting pellet, i.e., the “lysosomal fraction” contained a mixture of lysosomes (the dominant component), light mitochondria, peroxisomes, and endoplasmic reticulum. The pellet was suspended in 5 mL of 0.1% TRITON X-100 to release latent lysosomal enzymes. After that, the sample was frozen and stored at −20 °C until analysis. The samples were thawed, transferred to 1.5 mL Eppendorf tubes, and centrifuged at 12,000× *g* for 2 min to remove debris and insoluble material. The activity of β-galactosidase (BGAL, EC 3.2.1.23) was determined in lysosomal fraction according to Barrett’s method [29]. The protein level was determined by a modification of Lowry’s method [30] using bovine serum albumin as a standard. The activity of the enzyme was expressed in µmoles of product per mg of total protein per hour. Absorbance was measured with the use of a Spekol 1500 UV/VIS spectrophotometer (Analityk Jena AG). The results were subjected to statistical treatment with the use of XLSTAT add-in for Microsoft Excel. The conducted multi-way (age, genetic groups) analysis of variance ANOVA enabled the identification of significant effects of differences at the level of *p*-value. In addition, multiple pairwise comparisons were performed to analyze the difference between each pair using Dunn-Sidak’s test; *p* < 0.05 was considered a significance threshold.

*Plasma biomarkers*. Approximately 200–400 µL of peripheral blood was collected immediately after death by cardiac puncture. Blood from each animal was mixed with 3.2% of buffered sodium citrate in a 1.8 mL microtube (Becton Dickinson, Oxford, UK) and centrifuged at 4750× *g* for 10 min at room temperature. Next, plasma specimens were promptly collected and frozen at −70 °C. All further analyses were performed using the semi-automatic biochemical analyzer COBAS INTEGRA^®^ 400 plus system (Roche Diagnostics Ltd., Rotkreuz, Switzerland). All biochemical serum analytes, which are used to investigate main metabolic pathways and organ functions, were performed at the same time to minimize analytical variability. We analyzed selected parameters of blood: alanine transaminase (ALT), aspartate transaminase (AST), γ-glutamyl transferase (GGT), triacylglycerols (TAG), cholesterol (TCh); HDL-cholesterol (high-density fraction), lactate dehydrogenase (LDH), and creatinine. The code of all chemicals was inserted in equipment following the standards (Roche Diagnostics Ltd., Rotkreuz, Switzerland). The results were subjected to statistical treatment with the use of XLSTAT add-in for Microsoft Excel. The conducted multi-way (age, genetic groups) analysis of Kruskal–Wallis rank test (a nonparametric alternative to the ANOVA) enabled the identification of significant effects of differences at the level of the *p*-value. In addition, multiple pairwise comparisons were performed to analyze the difference between each pair using Dunn’s test; *p* < 0.05 was considered the significance threshold.

## 3. Results

### 3.1. Age-Related Changes in the Structure of Hepatocytes

The histology of liver cells was analyzed by light microscopy using semi-thin tissue sections. Adult (i.e., 7 months old) wild type mice showed normal cytoarchitecture of hepatic parenchyma (Figure 1A), which maintained a polygonal morphology with distinct cellular boundaries. Hepatocytes had a round vesicular nucleus and few lipid drops in the cytoplasm. No significant changes in liver structure were observed in the elderly (i.e., 17 months old) animals, with only a slight increased number of lipid droplets observed (Figure 1C and Figure 2). The adult ApoE^−/−^ mice had liver tissue with typical morphology (Figure 1B). However, a few differences between adult wild type and ApoE^−/−^ could be observed, because the latter ones had significantly greater steatosis (Figure 2). Moreover, the hepatocytes of elderly ApoE^−/−^ mice had moderate amounts of cytoplasm, vesicular cell nuclei, clear cell lines, and increased numbers of lipid drops compared to younger animals (Figure 1D and Figure 2). Furthermore, the number of lipid deposits observed in hepatocytes of elderly ApoE^−/−^ mice was markedly higher than that in hepatocytes of both younger ApoE^−/−^ mice and elderly wild type mice (Figure 2).

Hepatocytes from adult wild-type animals had “normal” ultrastructures of the nucleus, nucleolus, mitochondria, endoplasmic reticulum, and lysosomes (a representative micrograph is shown in Figure 3A). On the contrary, in hepatocytes from elderly mice, many damaged mitochondria with broken cristae, vacuolated matrixes, and a higher number of primary lysosomes were observed (Figure 3C,H,I and Figure 4A). The number of damaged mitochondria increased three-fold in hepatocytes of elderly mice when compared to hepatocytes of adult wild type animals (Figure 4B), yet no clear changes in the nucleus, nucleolus, and endoplasmic reticulum were observed. Ultrastructural changes observed between adult and elderly ApoE^−/−^ mice were more pronounced (Figure 3B,D–G). However, the length of mitochondria was larger in the adult and elderly ApoE^−/−^ mice compared to wild type mice (Figure 4C). Furthermore, nuclear lipid inclusions, lipofuscin deposits, and collagen fibers were observed in hepatocytes of elderly ApoE^−/−^ mice that were not found in adult ApoE^−/−^ animals (Figure 3B,D–G).

### 3.2. Age-Related Changes in the Structure of Cardiomyocytes

Semi-thin sections of heart from adult wild type and ApoE^−/−^ mice showed normal architecture of the myocardium. Myocytes were cylindrical, muscle fibers were in a parallel arrangement (Figure 5A,B). The heart tissue of elderly wild type mice showed mild swelling between the muscle fibers (Figure 5C). However, similar morphological changes were found in elderly Apo E^−/−^ mice (Figure 5D).

The ultrastructure of adult wild type and Apo E^−/−^ mice was comparable. Cell nuclei with a normal membrane and chromatin, as well as mitochondria with regular crest arrangements between myofibrils were observed in both cases (Figure 6A,B). Damaged mitochondria with swollen and broken cristae and broken inner membranes were observed in the cardiomyocytes of elderly mice, either wild type or ApoE^−/−^ (Figure 6C–E), and numbers were significantly increased when compared to adult animals (Figure 7A). The size of mitochondria increased in the cardiomyocytes of elderly animals from both strains. However, the extent of such changes was greater in the cardiomyocytes of ApoE^−/−^ mice (Figure 7B). Lipofuscin deposits were found at the nuclear pole of cardiomyocytes from elderly animals of both strains (Figure 6C,D). Moreover, numerous foam cells containing lipids were found in the blood vessels of elderly ApoE^−/−^ mice.

### 3.3. Age-Related Changes of Biochemical Markers

Several plasma biomarkers were tested in the blood of adult and elderly mice of both strains (Figure 8). The activity of ALT decreased in the plasma of elderly ApoE^−/−^ mice in comparison to adults, and was markedly lower than in plasma of elderly wild type animals. Moreover, levels of AST and GGT activity was generally lower in plasma of ApoE^−/−^ animals than in wild type. LDH was decreased in adult and elderly ApoE^−/−^ mice in comparison to elderly wild type mice. The level of triglycerides was lower in the plasma of ApoE^−/−^ animals, yet no age-related differences were observed within one strain. On the other hand, levels of cholesterol and HDL increased in the plasma of elderly ApoE^−/−^ mice and were markedly higher than levels in the plasma of elderly wild type animals. Additionally, the creatinine level significantly increased in the plasma of elderly wild type mice, while its level was constantly high in the plasma of either adult or elderly ApoE^−/−^ animals in comparison to wild type mice.

### 3.4. Analysis of Lysosomal Beta-Galactosidase

Furthermore, the activity of lysosomal β-galactosidase was analyzed in liver homogenates, which showed a significant increase in elderly animals from both strains of mice (Figure 9).

## 4. Discussion

Aging is associated with various morphological changes in the liver, however the mechanisms of these changes remain unclear [31]. Aging adversely affects liver function, cell viability, and tissue regeneration under pathological conditions caused, among others, by abnormal lipid metabolism associated with atherosclerosis [32]. Any abnormalities that occur with age in the liver are also reflected in heart function [33]. According to the work of Shah and Sass [26], liver damage known as “cardiac hepatopathy” is a common feature of variable hemodynamic disorders and can be associated with acute or chronic heart failure [34,35,36]. The detailed mechanism and biological function of cellular aging in liver disease has not been fully elucidated, although much research has been done in this field [1,18,37]. Cellular aging does not have a single marker, but it is mainly characterized by a combination of many markers, such as the expression of cell cycle inhibitors, β-galactosidase activity, and morphological changes such as an increase in hepatocyte size or the increase of polyploid hepatocytes in older mice [22,38].

Our research results focused on the assessment of morphological and ultrastructural changes in the liver and cardiac tissue in the elderly (17-month-old) mice, with either wild type or ApoE-deficient phenotypes. Markedly increased numbers of lipid deposits were observed in the livers of elderly ApoE-deficient animals, which was indicative of impaired lipid metabolism, yet no other significant changes were visible at the level of tissue morphology. However, several aging-related features were observed at the level of the ultrastructure of hepatocytes, which were more strongly expressed in ApoE-deficient animals. Increased numbers of damaged mitochondria were observed in elderly animals from either strain, yet aging-related changes were more profound in ApoE-deficient mice. It os noteworthy that quantitative and qualitative changes in mitochondria are observed in various kinds of aging cells [39,40]. Senescent mitochondria show structural deterioration, such as swelling, loss of cristae, and sometimes, the complete destruction of the inner membranes, resulting in the formation of amorphous electron-dense material [41]. Moreover, the presence of lipid drops, nuclear lipid inclusions, lipofuscin, and collagen fibers were noted in the hepatocytes of elderly ApoE-deficient animals. Accumulation of damaged macromolecules and organelles is one of the most persistent changes in aging cells and is associated with the decline of different catabolic pathways. However, our ultrastructural studies did not confirm the intensification of autophagic processes in aging hepatocytes. We did not observe an increase in the number of autophagic vacuoles and/or secondary lysosomes, that would confirm the increase of lysosomal degradation processes. An increase in the number of primary lysosomes and the presence of well-developed rough endoplasmic reticulum near mitochondria was found, which may indicate that the cell was synthesizing protein properly. Nevertheless, in agreement with previous studies [25,42,43], an increased number of lysosomes and lysosomal β-galactosidase activity was noted in hepatocytes of elderly animals, yet no significant differences were found between wild type and ApoE-deficient mice of the same age.

The most pronounced age-related changes occur in long-lived post-mitotic cells, including cardiac myocytes. These cells are particularly susceptible to aging due to their intense oxygen metabolism and production of ROS. The incidence of cardiomyopathy and heart failure gradually increases with age [21]. Hence, it has been suggested that the progressive accumulation of defective mitochondria is associated with cardiac dysfunction in the elderly [44]. Cardiac disorders frequently affect the liver and liver disorders affect the heart, and both effects can be accelerated by additional pathological environmental factors [33]. Hence, liver damage known as “cardiac hepatopathy” is a common feature of variable hemodynamic disorders and may be associated with acute or chronic heart failure [26]. Our ultrastructural analysis showed an accumulation of damaged mitochondria in the cardiomyocytes of elderly animals. Moreover, the increased fusion and/or decreased fission of mitochondria that resulted in their increased size was stronger in the cardiomyocytes of elderly ApoE-deficient mice. It is noteworthy that fission regulates morphology, facilitates mitochondrial trafficking, and segregates the most seriously damaged mitochondria to preserve the health of the mitochondrial network [45,46]. Furthermore, our results did not show autophagy in cardiomyocytes, which usually has a cytoprotective function.

At the biochemical level, we showed an aging-related increase in the level of cholesterol and HDL in the plasma of elderly ApoE-deficient mice, which correlated with the presence of foam cells in the heart tissue of such animals. This observation confirms that the lack of ApoE leads to an increase in VLDL (very low density lipoprotein) and chylomicron residues in the plasma and an increase in foam cell formation in the vessels, and consequent atherosclerosis [47,48]. The increases in total cholesterol and triglycerides, transformation of the lipoprotein profile, reduction of ApoE receptor synthesis and number, reduction of lipolytic enzyme activity, and intensity of fatty acid oxidation are known major age-related changes in lipid metabolism in ApoE^−/−^ mice [48,49]. Our observation suggests additionally that the phenotype of hypercholesterolemic ApoE^−/−^ mice is associated with aging of the liver and heart tissue.

## Figures and Tables

**Figure 1 cells-10-00502-f001:**
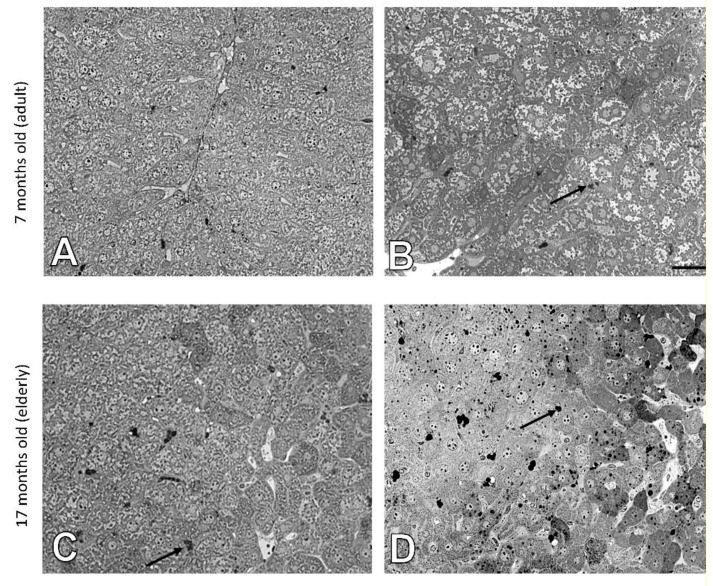
Age-related effects in the mouse liver. Semi-thin sections of mouse liver of 7-month-old (**A**) and 17-month-old C57Bl/6J mice (**C**); 7-month-old (**B**), and 17-month-old ApoE^−/−^ mice (**D**). Arrows indicate lipid droplets. Scale bar = 20 μm.

**Figure 2 cells-10-00502-f002:**
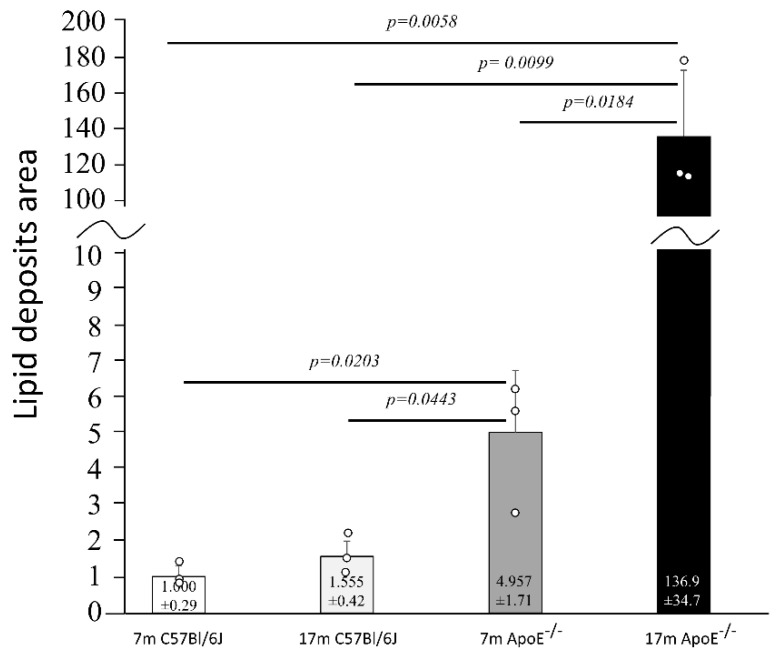
Senescence-related increase of the area of lipid deposits in liver cells. The relative area of lipid droplets in liver tissue of C57Bl/6J and ApoE^−/−^ mice is shown in relation to corresponding controls (7 months old). The actual area of lipid deposits in ApoE^−/−^ mice was larger than in C57Bl/6J mice; shown is the average number of pixels (pxs) attributed to lipid deposits normalized to a mean of 7m C57Bl/6J for visualization purposes. Data were analyzed by a two-way ANOVA test (the exact F values for genotype and age are 48.16 and 43.57, respectively), followed by Dunn–Sidak’s multiple comparison post-hoc test. Shown are the means ± standard deviations. The significance of differences between compared groups are marked with obtained *p*-values (*p*) for each significant comparison; the different genotype groups were not littermates; row data are depicted by circles; m–months.

**Figure 3 cells-10-00502-f003:**
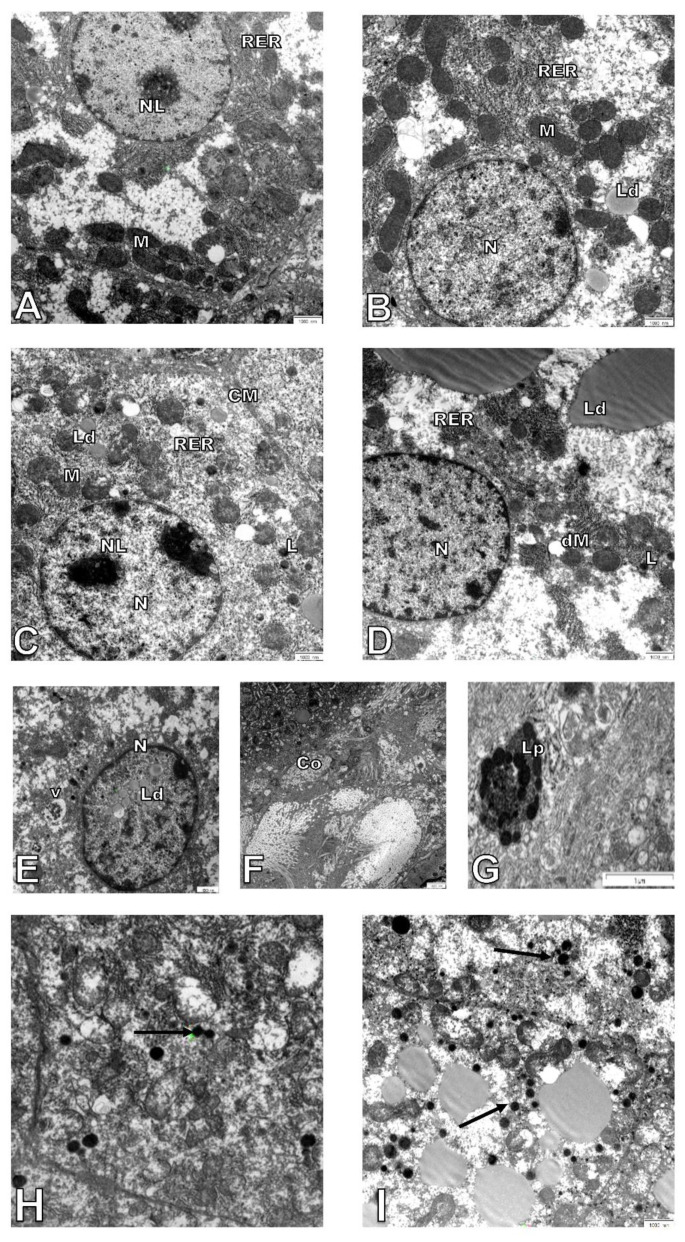
Age-related changes in hepatocyte ultrastructure. Seven-month-old (**A**) and 17-month-old C57Bl/6J mice (**C**,**H**); 7-month-old (**B**), and 17-month-old Apo E^−/−^ mice (**D**–**G**,**I**). CM—cell membrane, Lp—lipofuscin, Ld—lipid droplets, L—lysosomes, M—mitochondria, dM—damage mitochondria, N—nucleus, NL—nucleolus, RER—rough endoplasmic reticulum, V—vacuoles, Co—collagen. Arrows show lysosomes in older animals. Scale bar = 1000 nm.

**Figure 4 cells-10-00502-f004:**
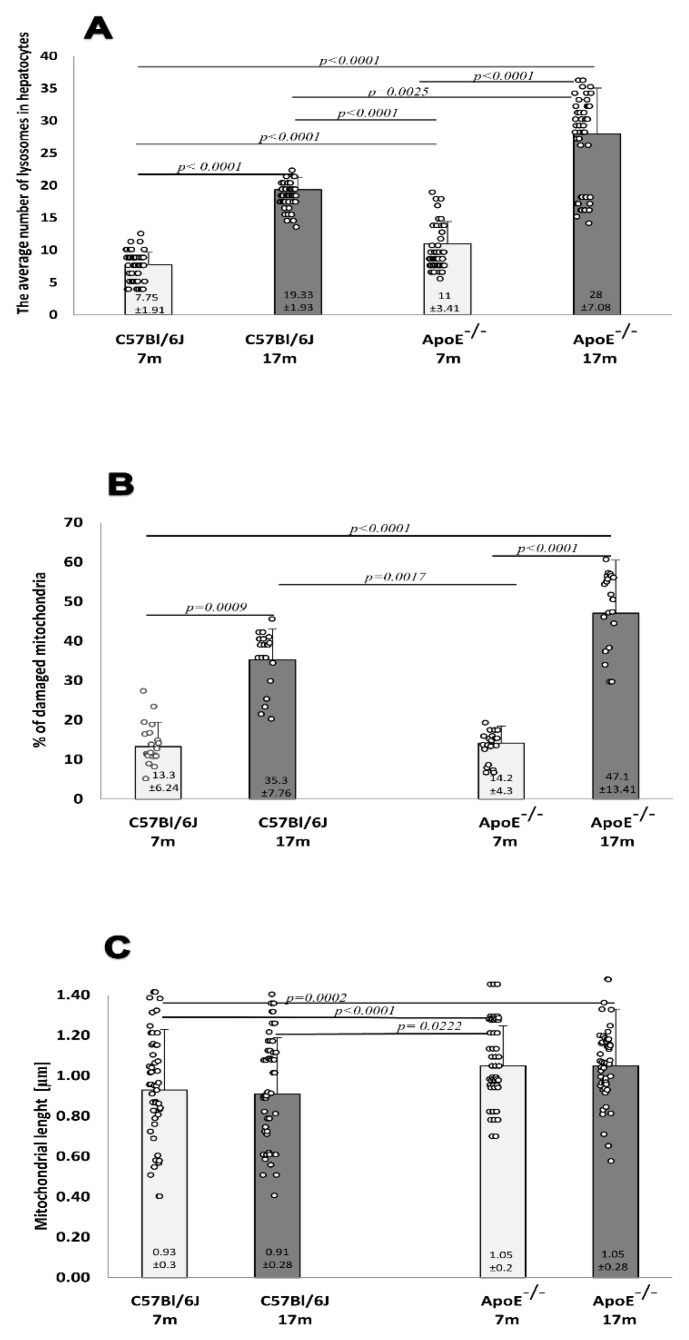
Senescence induces an increase in the number of damaged mitochondria and lysosomes in the mouse liver. Quantitative analysis of lysosomes in hepatocytes (**A**). The relative number of damaged mitochondria (percentage total per 1000 mitochondria counts) (**B**). The average length of mitochondria in hepatocytes (**C**) of C57Bl/6J and ApoE^−/−^ mice both aged 17 months and 7 months. Data were analyzed by a two-way Kruskal–Wallis test (exact F values for genotype and age in panel **A** are 113.79 and 562.74, in panel **B** are 5.41 and 101.54, and in panel **C** are 8.73 and 1.14, respectively) followed by Dunn’s multiple comparison post-hoc test. Shown are the means ± standard deviations. The significance of differences between compared groups are marked with obtained *p*-values (*p*) for each significant comparison; the different genotype groups were not littermates; row data are depicted by circles; m–months.

**Figure 5 cells-10-00502-f005:**
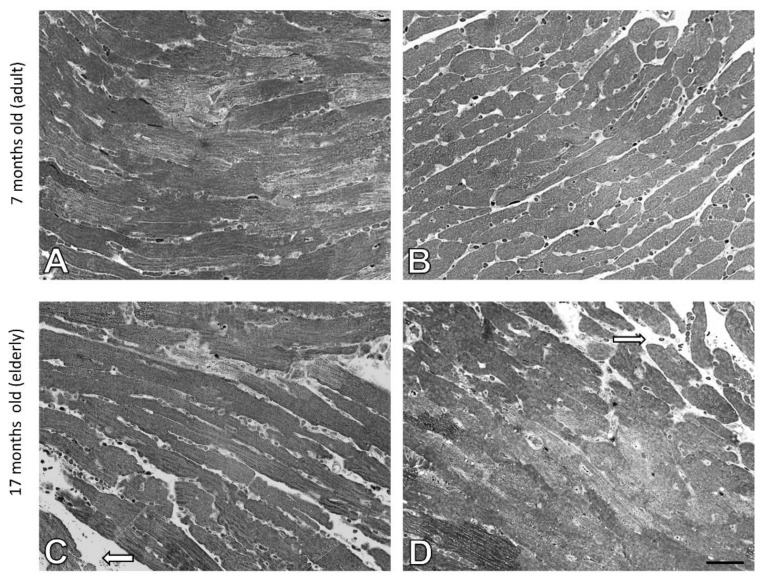
Swelling between the muscle fibers of the mice hearts. Semi-thin sections of mouse heart of 7-month-old (**A**) and 17-month-old C57Bl/6J mice (**C**); 7-month-old (**B**), and 17-month-old Apo E^−/−^ mice (**D**). The white arrow indicates mild edema. Scale bar = 20 μm.

**Figure 6 cells-10-00502-f006:**
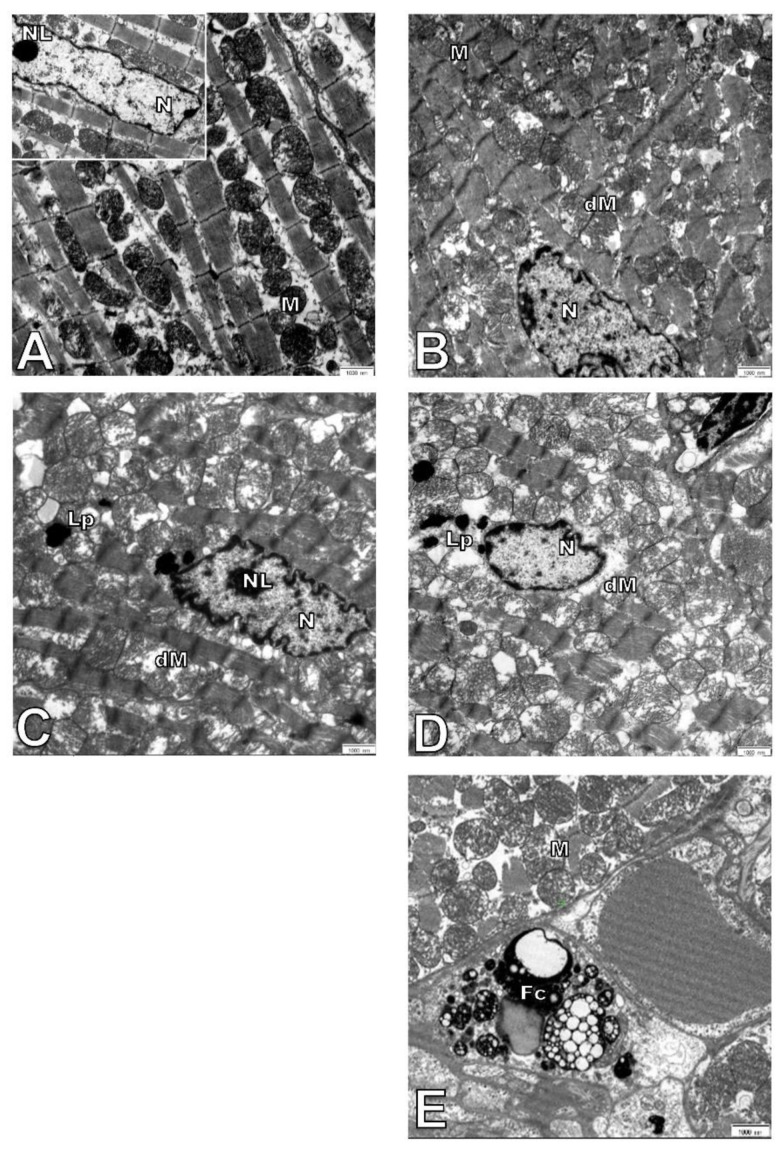
Electron micrographs showing ultrastructure of cardiomyocytes. The figure shows representative pictures of C57Bl/6J (**A**) and ApoE^−/−^ (**B**) 7-month-old mice and C57Bl/6J (**C**) and ApoE^−/−^ (**D**,**E**) 17-month-old mice. Fc—foam cell, Lp—lipofuscin, M—mitochondria, dM—damage mitochondria, N—nucleus, NL—nucleolus. Scale bar = 1000 nm.

**Figure 7 cells-10-00502-f007:**
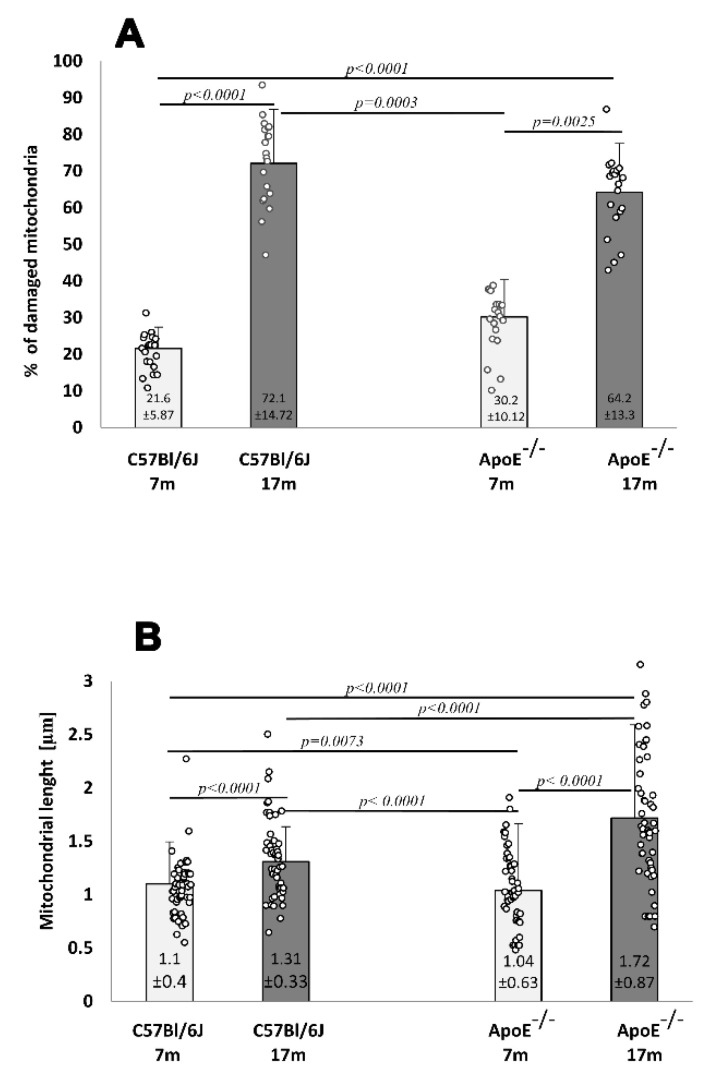
Senescence-related increases in damaged mitochondria and their length in the mouse hearts. The relative number of damaged cardiomyocyte mitochondria (percentage total 1000 mitochondria counts) (**A**). The average length of mitochondria in cardiomyocytes (**B**) of 7-month- and 17-month-old C57Bl/6J and ApoE^−/−^ mice. Data were analyzed by a two-way Kruskal–Wallis test (exact F values for genotype and age in panel **A** are 0.01 and 134.62 and in panel **B** are 10.08 and 70.38, respectively) followed by a Dunn’s multiple comparison post-hoc test. Shown are the means ± standard deviations. The significance of differences between compared groups are marked with obtained *p*-values (*p*) for each significant comparison; the different genotype groups were not littermates, row data are depicted by circles; m–months.

**Figure 8 cells-10-00502-f008:**
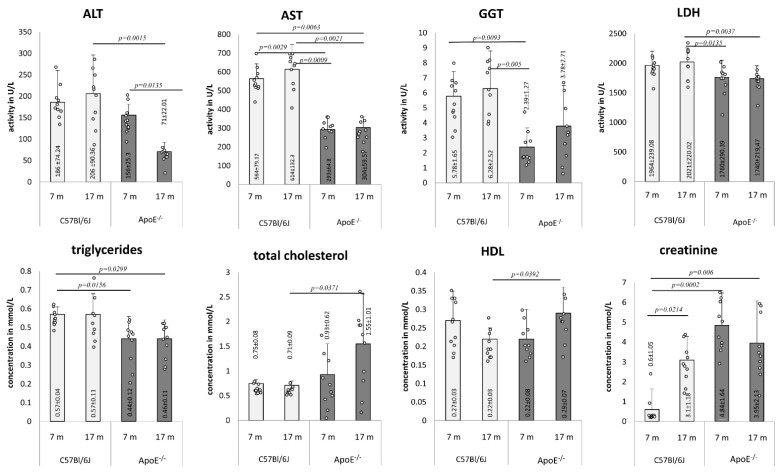
Age-related changes in mice blood parameters. Serum biochemical analysis in 7- and 17-month-old C57Bl/6J and ApoE^−/−^ mice. Data were analyzed by two-way Kruskal–Wallis test (the exact F values for genotype and age for alanine transaminase (ALT) are 4.86 and 0.43, for aspartate transaminase (AST) are 70.24 and 0.95, for γ-glutamyl transferase (GGT) are 13.48 and 1.54, for lactate dehydrogenase (LDH) are 11.98 and 3.97, for triglycerides are 9.72 and 0.03, for total cholesterol are 6.14 and 2.2, for HDL are 0.07 and 0.07, and for creatinine are 19.21 and 1.66, respectively) followed by Dunn’s multiple comparison post-hoc test. Shown are the means ± standard deviations. The significance of differences between compared groups are marked with obtained *p*-values (*p*) for each significant comparison; the different genotype groups were not littermates; row data are depicted by circles; m–months.

**Figure 9 cells-10-00502-f009:**
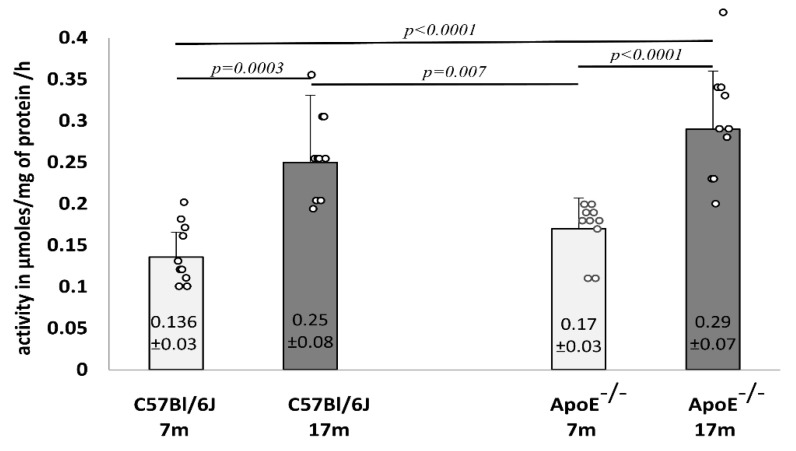
Senescence-related increase in the activity of β-galactosidase in the liver tissue of 7-month-old and 17-month-old C57Bl/6J and ApoE^−/−^ mice. Data were analyzed by a two-way ANOVA test (exact F values for genotype and age are 3.26 and 34.27, respectively) followed by a Dunn–Sidak’s multiple comparison post-hoc test. Shown are the means ± standard deviations. The significance of differences between compared groups are marked with obtained *p*-values (*p*) for each significant comparison, the different genotype groups were not littermates; row data are depicted by circles; m-months.

## Data Availability

Data presented in this study are contained in this article, or available upon request to the corresponding autors.

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
