# Peer review of "Aging-Related Changes in the Ultrastructure of Hepatocytes and Cardiomyocytes of Elderly Mice Are Enhanced in ApoE-Deficient Animals"

_cells, 2021, doi:10.3390/cells10030502_

Round 1

Reviewer 1 Report

In the manuscript “Aging-related changes in the ultrastructure of hepatocytes and cardiomyocytes of elderly mice are enhanced in ApoE-deficient animals”, the authors study the impact of an impaired lipid metabolism in promoting age-related deterioration of liver and heart tissues. In order to model lipid disorders, the authors adopt and ApoE -/- mouse strain which is one of the most widely used model for the study of age-related metabolic impairment and cardiovascular disorders. Ageing is evaluated by combining characterization of ultrastructural (number of damaged mitochondria, ER and nucleus damage), morphological (cellular swelling) and functional (b-Gal) changes at cellular and intracellular levels with the quantitation of some blood biomarkers (including creatinine, triglycerides, cholesterol, alanine transaminase, glutamyl transferase, etc)

In general, the research presented in the manuscript is well organized although I would like to encourage the authors to address to some issues, I believe the manuscript presents.

Methods Section

  1. The authors did not indicate the exact number of animals that were employed in the study. That should be included in the first paragraph of the method section as well as in addition to explanation of statistical approaches adopted for the analysis of the data (with particular reference to biochemical markers data).
  2. The mice included in the study were only male so it cannot be evaluated whether this effect is sex-driven. Could the authors specify whether they expect to observe sex-related, tissue-specific differences in age-related decline of hyperlipidemic mice?
  3. The authors should specify the statistical methods adopted for the study of age-related changes in plasma biomarkers.

Results Section

  1. In Figures 2, 4, 7 and 8 the authors should mention the number of slides/samples they performed the analysis on.
  2. In the paragraph “Age-related changes of biochemical markers” the author did not include any pictures related to the evaluation of age-related changes of plasma biomarkers as they would be an additional information supporting the author’s hypothesis.

Overall Comments

  1. The evaluation of cellular ageing through the study of structural changes has been rather neglected in comparison to the analysis of molecular markers, as the authors mentioned, including the expression of cell-cycle regulators (ie. p16 and p21), the accumulation of DNA damage-related marks (ie. gH2AX) or the increased production of oxygen reactive species (ROS). I would encourage the author to include one of these characterization (ie through immunohistochemistry) since it would provide molecular confirmation (while functional and morphological ones have already been shown) that a dysfunctional lipid metabolism leading to hyperlipidemia is indeed linked to accelerated age-related tissue deterioration.
    Similarly at systemic level, the authors provided several information regarding plasma biomarkers of age while neglecting other more studied biomarkers such as PBMC DNA methylation. Nevertheless, since the manuscript focuses heavily on evaluation of age-related deterioration of cardiac and hepatic tissues, I would suggest to focus firstly on tissue level before moving to a systemic one.

Author Response

The authors would like to thank reviewer #1 for his constructive comments on our manuscript. All suggestions were considered.

In the manuscript “Aging-related changes in the ultrastructure of hepatocytes and cardiomyocytes of elderly mice are enhanced in ApoE-deficient animals”, the authors study the impact of an impaired lipid metabolism in promoting age-related deterioration of liver and heart tissues. In order to model lipid disorders, the authors adopt and ApoE -/- mouse strain which is one of the most widely used model for the study of age-related metabolic impairment and cardiovascular disorders. Ageing is evaluated by combining characterization of ultrastructural (number of damaged mitochondria, ER and nucleus damage), morphological (cellular swelling) and functional (b-Gal) changes at cellular and intracellular levels with the quantitation of some blood biomarkers (including creatinine, triglycerides, cholesterol, alanine transaminase, glutamyl transferase, etc)

In general, the research presented in the manuscript is well organized although I would like to encourage the authors to address to some issues, I believe the manuscript presents.

Methods Section

  1. The authors did not indicate the exact number of animals that were employed in the study. That should be included in the first paragraph of the method section as well as in addition to explanation of statistical approaches adopted for the analysis of the data (with particular reference to biochemical markers data).

Text is revised and information is added:

Four mice from each test group were used to evaluate morphological analysis. Further, 12 epon blocks were prepared for each individual group of mice. The ultrastructural changes was performed on three grids per resin block. For histology, 10 specimens of 10 semi-thin sections were prepared and 100 pictures were taken in each group of mice.

The following information about statistical approaches has been included in the Methods of the revised manuscript:

Analysis of ultrastructure.

The results were subjected to statistical treatment with the use of XLSTAT add-in for Microsoft Excel. The conducted multi-way (age, genetics groups) analysis of Kruskal-Wallis rank test (nonparametric alternative to the ANOVA) enabled the identification of significant effects of differences at the level of p–value. In addition, multiple pairwise comparisons were performed to analyze the difference between each pair using Dunn's test; p<0.05 was considered a significance threshold.

Analysis of histomorphology. 10 specimens of 10 semi-thin sections were prepared and 100 pictures were taken in each group of mice. The results were subjected to statistical treatment with the use of XLSTAT add-in for Microsoft Excel. The conducted multi-way (age, genetics groups) analysis of variance ANOVA enabled the identification of significant effects of differences at the level of p–value. In addition, multiple pairwise comparisons were performed to analyze the difference between each pair using Dunn-Sidak's test; p<0.05 was considered a significance threshold.

Plasma biomarkers. The results were subjected to statistical treatment with the use of XLSTAT add-in for Microsoft Excel. The conducted multi-way (age, genetics groups) analysis of Kruskal-Wallis rank test (nonparametric alternative to the ANOVA) enabled the identification of significant effects of differences at the level of p–value. In addition, multiple pairwise comparisons were performed to analyze the difference between each pair using Dunn's test; p<0.05 was considered a significance threshold.

  1. The mice included in the study were only male so it cannot be evaluated whether this effect is sex-driven. Could the authors specify whether they expect to observe sex-related, tissue-specific differences in age-related decline of hyperlipidemic mice?

Thank you for raising this important point.

The mice used in the presented study were included in the large multicenter project Cardiorisk. All mice (wild type and ApoE-deficient were purchased from Charles River Laboratories (Research Models and Services, Germany GmbH). Only male mice were used in the study. Presented project was aimed to evaluate the animals at the late time point to analyse the senescence induced changes not the gender.

It was shown by the others that there are sex-related changes in the mice in respect to wild type and hyperlipidemic mice. This could be the topic of the future project

The information is added:

Male C57Bl/6J mice, either wild-type (wt) or homozygous ApoE deficient (ApoE-/-) were purchased from Charles River Laboratories (Research Models and Services, Germany GmbH); the different genotype groups were not littermates. In each group we included 10 animals.

  1. The authors should specify the statistical methods adopted for the study of age-related changes in plasma biomarkers.

Text is revised, statistical methods has been rewritten and additional information is added in the revised manuscript.

Analysis of ultrastructure.

The results were subjected to statistical treatment with the use of XLSTAT add-in for Microsoft Excel. The conducted multi-way (age, genetics groups) analysis of Kruskal-Wallis rank test (nonparametric alternative to the ANOVA) enabled the identification of significant effects of differences at the level of p–value. In addition, multiple pairwise comparisons were performed to analyze the difference between each pair using Dunn's test; p<0.05 was considered a significance threshold.

Analysis of histomorphology. 10 specimens of 10 semi-thin sections were prepared and 100 pictures were taken in each group of mice. The results were subjected to statistical treatment with the use of XLSTAT add-in for Microsoft Excel. The conducted multi-way (age, genetics groups) analysis of variance ANOVA enabled the identification of significant effects of differences at the level of p–value. In addition, multiple pairwise comparisons were performed to analyze the difference between each pair using Dunn-Sidak's test; p<0.05 was considered a significance threshold.

Plasma biomarkers. The results were subjected to statistical treatment with the use of XLSTAT add-in for Microsoft Excel. The conducted multi-way (age, genetics groups) analysis of Kruskal-Wallis rank test (nonparametric alternative to the ANOVA) enabled the identification of significant effects of differences at the level of p–value. In addition, multiple pairwise comparisons were performed to analyze the difference between each pair using Dunn's test; p<0.05 was considered a significance threshold.

Results Section

  1. In Figures 2, 4, 7 and 8 the authors should mention the number of slides/samples they performed the analysis on.

We thank Reviewer#1 for these comments. We add additional information in material and methods part.

Three mice from each test group were used to evaluate morphological changes at the tissue and cellular levels. Further, 3 epon blocks were prepared for each individual. For histology, 10 specimens of 10 semi-thin sections were prepared and 100 pictures were taken in each group of mice. The ultrastructural analysis was performed on the basis of 3 ultra-thin sections per individual from a given group (9 sections per group of animals).

  1. In the paragraph “Age-related changes of biochemical markers” the author did not include any pictures related to the evaluation of age-related changes of plasma biomarkers as they would be an additional information supporting the author’s hypothesis.

In our project, we decided to evaluate the changes in the liver and heart (Figure 1 and 5) of both groups of mice and then further evaluate the changes in hepatocytes and cardiomyocytes ultrastructure (Figure 3 and 6). Biochemical markers from plasma were evaluated to check that there is an age-related change and the type of mouse is important (Figure 8). We agree with the reviewer that adding immunohistochemistry studies could tell us more about the changes that are present, but during the tissue fixation procedure, we used antigenicity-suppressing fixers (glutaraldehyde and osmium tetroxide), and we embedded the tissues in epoxy resin, which is impenetrable for antibodies. The reagents used by us retain the cell structure very well, unfortunately they make it impossible to carry out immunocytochemical reactions (we did not plan them in the tissue processing phase), and we do not have additional samples for such analysis.

Overall Comments

  1. The evaluation of cellular ageing through the study of structural changes has been rather neglected in comparison to the analysis of molecular markers, as the authors mentioned, including the expression of cell-cycle regulators (ie. p16 and p21), the accumulation of DNA damage-related marks (ie. gH2AX) or the increased production of oxygen reactive species (ROS). I would encourage the author to include one of these characterization (ie through immunohistochemistry) since it would provide molecular confirmation (while functional and morphological ones have already been shown) that a dysfunctional lipid metabolism leading to hyperlipidemia is indeed linked to accelerated age-related tissue deterioration.

Similarly, at systemic level, the authors provided several information regarding plasma biomarkers of age while neglecting other more studied biomarkers such as PBMC DNA methylation. Nevertheless, since the manuscript focuses heavily on evaluation of age-related deterioration of cardiac and hepatic tissues, I would suggest to focus firstly on tissue level before moving to a systemic one.

We thank Reviewer#1 for these important comments.

In our project, we decided to evaluate the changes in the liver and heart (Figure 1 and 5) of both groups of mice and then further evaluate the changes in hepatocytes and cardiomyocytes ultrastructure (Figure 3 and 6). Biochemical markers from plasma were evaluated to check that there is an age-related change and the type of mouse is important (Figure 8). We agree with the reviewer that adding immunohistochemistry studies could tell us more about the changes that are present, but during the tissue fixation procedure, we used antigenicity-suppressing fixers (glutaraldehyde and osmium tetroxide), and we embedded the tissues in epoxy resin, which is impenetrable for antibodies. The reagents used by us retain the cell structure very well, unfortunately they make it impossible to carry out immunocytochemical reactions (we did not plan them in the tissue processing phase), and we do not have additional samples for such analysis.

On behalf of my co-authors, I once again express my sincere thanks to the erudite reviewers for the valuable suggestions and constructive input to improve the quality of our manuscript.

Reviewer 2 Report

All figures: please overlay raw datapoints onto bargraphs, as solid bar graphs are an inappropriate way to show data as they can obscure true data distribution. Please also specify in figure legends what error bars represent: SEM, SD, IQ etc? I would suggest using interquartile range or SD, but not SEM. Please also list n values in each figure legend, and clarify whether this represents the number of animals or number of SEM sections or a combination of both. Exact p-values should be listed. 

It is not clear how data were analysed. In the methods section a Mann-Whitney U test is listed, however this is only appropriate for comparing two groups. In figures 2, 4 and 7 there are multiple comparisons made within the same figure, therefore a correction for multiple comparisons should be used. I would strongly suggest that data should be analysed by 2-way ANOVA, using age and genotype as the two interaction terms, followed by some kind of post-hoc multiple comparison test (e.g. Sidak's test). 

Labelling of figures can be vastly improved, e.g. Fig 7b y-axis title: "The length of mitochondria in uM" could be changed to "Mitochondrial length (uM)"

Please clarify in the methods section whether WT mice and ApoE null mice were littermates? This should be the case in order to make direct comparisons: heterozygous ApoE mice should be bred, and WT/KO littermates used for direct comparison. If they were obtained from separate colonies, then no direct comparisons should be made between the two genotypes, as is currently the case - or, the manuscript and the figure legends should clearly state that this huge caveat. 

Page 11: change "heart hepatopathy" to either "cardiac hepatopathy" or "congestive hepatopathy"

Data in fig. 8 does not seem to match description of plasma biomarkers on page 10? Fig legend describes SA-beta gal activity. Ignoring the description of plasma biomarkers in Fig 8 it is not clear what is actually being measured in this graph: y-axis is labelled as "uM per mg protein / hour" - uM of what per mg? What is being measured?

Author Response

Reviewer #2

The authors would like to thank reviewer #2 for his evaluation of our work and his comments on our manuscript. All changes in the text based on the comments of reviewer #2 are considered.

  1. All figures: please overlay raw datapoints onto bargraphs, as solid bar graphs are an inappropriate way to show data as they can obscure true data distribution. Please also specify in figure legends what error bars represent: SEM, SD, IQ etc? I would suggest using interquartile range or SD, but not SEM. Please also list n values in each figure legend, and clarify whether this represents the number of animals or number of SEM sections or a combination of both. Exact p-values should be listed.

We thank Reviewer#2 for these comments

Figures are corrected and figures legend is revised and information as SD and p-values are added:

Shown are means ± standard deviations. The significance of differences between compared groups are marked with obtained p-values (p) for each significant comparison; the different genotype groups were not littermates; m–months.

The information about number animals/section is added in the material and methods part in revised manuscript:

Four mice from each test group were used to evaluate morphological analysis. Further, 12 epon blocks were prepared for each individual group of mice. The ultrastructural changes was performed on three grids per resin block. For histology, 10 specimens of 10 semi-thin sections were prepared and 100 pictures were taken in each group of mice.

  1. It is not clear how data were analysed. In the methods section a Mann-Whitney U test is listed, however this is only appropriate for comparing two groups. In figures 2, 4 and 7 there are multiple comparisons made within the same figure, therefore a correction for multiple comparisons should be used. I would strongly suggest that data should be analysed by 2-way ANOVA, using age and genotype as the two interaction terms, followed by some kind of post-hoc multiple comparison test (e.g. Sidak's test). 

We agree with the reviewer’s comment. Some of the subsections in the Methods section were lacking the description of the chosen statistical method, therefore the adequate description was added to each subsection where the statistical analysis was performed. Regarding the multiple comparison issue we also have to admit that Student’s t-test and Mann-Whitney test may not be optimal choice in case of comparison of more than two groups. Therefore, we recalculated all the statistical analysis and instead of using Student’s t-test we used 2-way ANOVA with Dunn-Sidak's post-hoc tests for multiple pairwise comparisons (in case of data with normal distribution) and instead of Mann-Whitney U test we used Kruskal-Wallis rank test with Dunn's post-hoc tests for multiple pairwise comparisons (in case of data that did not follow the normal distribution).

The following information about statistical approaches has been included in the Methods of the revised manuscript:

Analysis of ultrastructure.

The results were subjected to statistical treatment with the use of XLSTAT add-in for Microsoft Excel. The conducted multi-way (age, genetics groups) analysis of Kruskal-Wallis rank test (nonparametric alternative to the ANOVA) enabled the identification of significant effects of differences at the level of p–value. In addition, multiple pairwise comparisons were performed to analyze the difference between each pair using Dunn's test; p<0.05 was considered a significance threshold.

Analysis of histomorphology. 10 specimens of 10 semi-thin sections were prepared and 100 pictures were taken in each group of mice. The results were subjected to statistical treatment with the use of XLSTAT add-in for Microsoft Excel. The conducted multi-way (age, genetics groups) analysis of variance ANOVA enabled the identification of significant effects of differences at the level of p–value. In addition, multiple pairwise comparisons were performed to analyze the difference between each pair using Dunn-Sidak's test; p<0.05 was considered a significance threshold.

Plasma biomarkers. The results were subjected to statistical treatment with the use of XLSTAT add-in for Microsoft Excel. The conducted multi-way (age, genetics groups) analysis of Kruskal-Wallis rank test (nonparametric alternative to the ANOVA) enabled the identification of significant effects of differences at the level of p–value. In addition, multiple pairwise comparisons were performed to analyze the difference between each pair using Dunn's test; p<0.05 was considered a significance threshold.

  1. Labelling of figures can be vastly improved, e.g. Fig 7b y-axis title: "The length of mitochondria in uM" could be changed to "Mitochondrial length (uM)"

We have revised the Figure 7 legend in the revised manuscript

  1. Please clarify in the methods section whether WT mice and ApoE null mice were littermates? This should be the case in order to make direct comparisons: heterozygous ApoE mice should be bred, and WT/KO littermates used for direct comparison. If they were obtained from separate colonies, then no direct comparisons should be made between the two genotypes, as is currently the case - or, the manuscript and the figure legends should clearly state that this huge caveat. 

We agree with the reviewer’s comment. The breeding scheme was not mentioned in the Methods. We updated the information that the different genotype groups were not littermates in the Methods section and under each Figure where the genotypes were directly compared.

  1. Page 11: change "heart hepatopathy" to either "cardiac hepatopathy" or "congestive hepatopathy"

Text is revised and “heart hepatopathy’’ is replaced with “cardiac hepatopathy’’.

  1. Data in fig. 8 does not seem to match description of plasma biomarkers on page 10? Fig legend describes SA-beta gal activity. Ignoring the description of plasma biomarkers in Fig 8 it is not clear what is actually being measured in this graph: y-axis is labelled as "uM per mg protein / hour" - uM of what per mg? What is being measured?

We rewrite the part on plasma biomarkers in revised manuscript:

Age-related changes of biochemical markers

Several plasma biomarkers were tested in the blood of adult and elderly mice of both strains (Figure 8). The activity of ALT decreased in plasma of elderly ApoE-/- mice in comparison to adult and was markedly lower than in plasma of elderly wild type animals. Moreover, levels of AST and GGT activity was generally lower in plasma of ApoE-/- animals than in wild type. LDH was decreased in adult and elderly ApoE-/- in comparison to elderly wild type. The level of triglycerides was lower in plasma of ApoE-/- animals, yet no age-related differences were observed within one strain. On the other hand, levels of cholesterol and HDL increased in plasma of elderly ApoE-/- mice and were markedly higher than in plasma of elderly wild type animals. Additionally, the creatinine level significantly increased in plasma of elderly wild type mice while its level was constantly high in plasma of either adult or elderly ApoE-/- animals in comparison to wild type mice.

Analysis of lysosomal beta-galactosidase

Furthermore, the activity of lysosomal β-galactosidase was analyzed in liver homogenates, which showed a significant increase in elderly animals from both strains of mice (Figure 9).

The axis has been rewritten and clear label was added: activity in µmoles/mg of protein/hour.

On behalf of my co-authors, I once again express my sincere thanks to the erudite reviewers for the valuable suggestions and constructive input to improve the quality of our manuscript.

Round 2

Reviewer 1 Report

I reviewed the changes the authors made to their manuscript. 
I believe the authors did an excellent job addressing the points raised therefore enriching the manuscript for a clearer comprehension of the changes affecting liver and heart upon ageing in hyperlipidemic conditions.

My suggestion regarding the inclusion of immunohystochemistry of senescence-related markers was aimed to raise the need for a molecular validation of tissue-specific ageing decline. I understand the technical reasons the authors indicated in their reply.

Author Response

Manuscript Ref. No: cells-1107760

Title: Aging-related changes in the ultrastructure of hepatocytes and cardiomyocytes of elderly mice are enhanced in ApoE-deficient animals

Authors: Malgorzata Lysek-Gladysinska *, Anna Wieczorek, Artur Jóźwik *, Anna Walaszczyk, Karol Jelonek, Grażyna Szczukiewicz-Markowska, Olaf Horbańczuk, Monika Pietrowska, Piotr Widlak, Dorota Gabryś

We thank you very much for the careful revision to our manuscript. 

Answers to Reviewers' Comments

Query:

I reviewed the changes the authors made to their manuscript. 
I believe the authors did an excellent job addressing the points raised therefore enriching the manuscript for a clearer comprehension of the changes affecting liver and heart upon ageing in hyperlipidemic conditions.

My suggestion regarding the inclusion of immunohistochemistry of senescence-related markers was aimed to raise the need for a molecular validation of tissue-specific ageing decline. I understand the technical reasons the authors indicated in their reply.

Reply: We thank you very much for the kind and professional review of our manuscript.

Thanks to these comments, this manuscript acquired more scientific value.

We are also very grateful to the Reviewer for your understanding that we could not perform more immunohistochemistry analysis.

Such more detailed studies are not feasible within the current study, but we do agree with the Reviewer that this could be a highly interesting topic for future investigation.

Thanks again for your comments.

On behalf of my co-authors, I once again express my sincere thanks to the erudite reviewers for the valuable suggestions and constructive input to improve the quality of our manuscript.

Kind regards,

Artur Jóźwik

Reviewer 2 Report

Thank you for addressing these comments, however graphs still do not show raw data points - these must be overlaid onto bar graphs. 

In each figure legend, it is still not clear which test was used - for example I presume it should say "data were analysed by 2-way ANOVA followed by Dunn-Sidak multiple comparison test". If you are interested in the effect of each factor (e.g. age, genotype) rather than the combination of both then I would also show the exact F value for the ANOVA effect. 

Author Response

Manuscript Ref. No: cells-1107760

Title: Aging-related changes in the ultrastructure of hepatocytes and cardiomyocytes of elderly mice are enhanced in ApoE-deficient animals

Authors: Malgorzata Lysek-Gladysinska *, Anna Wieczorek, Artur Jóźwik *, Anna Walaszczyk, Karol Jelonek, Grażyna Szczukiewicz-Markowska, Olaf Horbańczuk, Monika Pietrowska, Piotr Widlak, Dorota Gabryś

Answers to Reviewers' Comments

We thank you very much for the careful revision to our manuscript. We realized from the comments received that several key points in our original work have not been properly addressed. Additionally, we also recognized and appreciated the competence of the Reviewers. We have incorporated most of the suggestions of the reviewers that certainly improved the quality of our manuscript

Reviewers' comments:

Reviewer #2

The authors would like to thank reviewer 2 for his evaluation of our work and his comments on our manuscript. All changes in the text based on the comments of reviewer 2 are considered.

Query 1:

Thank you for addressing these comments, however graphs still do not show raw data points - these must be overlaid onto bar graphs. 

Reply 1: We thank the Reviewer for the detailed assessment of our revised manuscript. We did our best to address all the remaining issues raised by the reviewer.

Thank you for your attention on these charts. It was our oversight, now the raw results have been plotted on the bars as points.

Query 2:

In each figure legend, it is still not clear which test was used - for example I presume it should say "data were analysed by 2-way ANOVA followed by Dunn-Sidak multiple comparison test". If you are interested in the effect of each factor (e.g. age, genotype) rather than the combination of both then I would also show the exact F value for the ANOVA effect.

Reply 2: We thank the Reviewer for this excellent suggestion and we have added F value which includes effect of each factors (age and genotypes).

Thanks again for your comments.

On behalf of my co-authors, I once again express my sincere thanks to the erudite reviewers for the valuable suggestions and constructive input to improve the quality of our manuscript.

Kind regards,

Artur Jóźwik